# STAT1 as a tool for non-invasive monitoring of NK cell activation in cancer
Jin Young Min[1,7], Hye Min Kim [1,2,7], Hyunseung Lee[1], Mi Young Cho[1], Hye Sun Park [1], Sang-Yeop Lee[1,2], Min Sung Park[1,2], Sang Keun Ha[2,3], Donghwan Kim[3], Hye Gwang Jeong[4], Tae-Don Kim [2,5], Kwan Soo Hong[1,6,8] ✉ & Eun Hee Han [1,2,8] ✉

Natural killer (NK) cells play a crucial role in immunotherapy for cancer due to their natural ability to target and destroy cancer cells. However, current methods to visualize NK cells' activity against tumors in live organisms are limited. We introduce an imaging method that non-invasively tracks NK cell activation by cancer cells through the STAT1 protein. To achieve this, we modified NK cells to include a specific genetic sequence that binds to STAT1 when activated. These engineered NK cells (GAS-NK) demonstrate their functionality through various biological tests and analysis. Observations of changes in cancer environments and patient-derived cancer organoid models further confirm the effectiveness of this approach. Our method provides a way to monitor NK cell activity, which could improve the prediction and effectiveness of NK cell-based cancer therapies, contributing to advances in cancer treatment.

Natural killer (NK) cells, integral components of the innate immune system, play a pivotal role in the host's defense mechanisms against both viral infections and cancer proliferation[1,2]. Extensive research has focused on unraveling the antitumor properties of NK cells, exploring facets such as immune surveillance, evasion strategies employed by malignant cells, and potential adverse effects[3]. A significant challenge in advancing NK cell therapy lies in selecting treatments with optimal anticancer efficacy, compounded by an incomplete understanding of the biological factors influencing the success of NK cell therapy in tumor patients[3,4]. The progress of NK cell therapy critically depends on the ability to quantify the anticancer capability of NK cells and selectively identify cells within the administered population exhibiting the highest efficacy for patient treatment.

The development of imaging technology for direct and real-time monitoring NK cell activation in vivo during NK cell therapy administration has become increasingly crucial[5,6]. Advances in non-invasive imaging modalities, including magnetic resonance (MR) imaging, optical imaging, fluorodeoxyglucose positron emission tomography (FDG-PET), and various radiotracer methods, are pivotal in ongoing efforts to develop and evaluate cancer immunotherapies, providing insights into NK cell biodistribution[5,6]. However, a limitation of existing imaging technology is its inability to distinguish among NK cell populations with varying anticancer effects, imaging and monitoring all administered NK

cells simultaneously. Therefore, the need for nonclinical imaging techniques capable of discerning NK cell populations based on their anticancer efficacy is essential for informed decision-making before administering NK cell therapy to patients.

Considering that only a limited subset of cancer patients responds favorably to NK cell therapy, there is a pressing need for reliable biomarkers to effectively identify patients likely to benefit from this treatment[7,8]. Despite extensive efforts to identify biomarkers, such as checkpoint ligand expression, mutational burden, neoantigen expression, interferon signatures, and characteristics of the inflammatory tumor microenvironment, definitive predictors of patient response to immunotherapy remain elusive[9–11]. Responsive tumors often exhibit gene expression profiles characterized by enhanced signal transducer and activator of transcription 1 (STAT1) and toll-like receptor 3 (TLR3) signaling, coupled with a concurrent decrease in interleukin-10 (IL-10) expression[12]. Nevertheless, direct markers accurately assessing patient responsiveness to NK cell therapy are continuously needed[13]. NK cells produce a variety of cytokines, such as interferon-gamma (IFNγ) and tumor necrosis factor-alpha (TNF-α), which have direct antitumor effects and can also enhance the adaptive immune response against tumors[12]. IFNγ, in particular, plays a critical role in enhancing the antigen presentation process and activating macrophages[8]. Imaging NK cells with high anticancer efficacy selectively within the administered NK cell

[1]Biopharmaceutical Research Center, Korea Basic Science Institute (KBSI), Cheongju, 28119, Republic of Korea. [2]Korea University of Science and Technology (UST), Daejeon, 34113, Republic of Korea. [3]Food Functionality Research Division, Korea Food Research Institute, Jeollabuk-do, 55365, Republic of Korea. [4]College of Pharmacy, Chungnam National University, Daejeon, 34134, Republic of Korea. [5]Korea Research Institute of Bioscience and Biotechnology, Daejeon, 34141, Republic of Korea. [6]Department of Chemistry, Chung-Ang University, Seoul, 06974, Republic of Korea. [7]These authors contributed equally: Jin Young Min, Hye Min Kim. [8]These authors jointly supervised this work: Kwan Soo Hong, Eun Hee Han. ✉e-mail: kshong@kbsi.re.kr; heh4285@kbsi.re.kr

population at the individual level can serve as predictive markers, informing other treatment strategies.

In this study, we present a approach for selectively monitoring activated NK cells. Using a pGF-GAS lentivirus, we engineered NK cells to incorporate a reporter gene capable of producing fluorescence and luminescence upon activation, enabling the selective monitoring of STAT1-dependent NK cell activation. STAT1 is a transcriptional regulator critical in the production of effector molecules upon NK cell activation. We rigorously validated this methodology by co-culturing these engineered NK cells (GAS-NK) with a diverse range of normal cells, various cancer cell lines, animal models, and cancer organoids. Our study provides not only a comprehensive assessment of the potential of GAS-NK cells but also investigates the activation dynamics of GAS-NK cells, as indicated by cytokine expression patterns in cancer cell culture media. This approach holds promise in utilizing GAS-NK cell activation as a biomarker to differentiate responders from non-responders to NK cell therapy, potentially aiding in predicting the efficacy of NK cell-based cancer therapies and screening effective NK cell therapy candidates.

## Results

### Selective monitoring of activated GAS-NK cells in vitro and in vivo

NK cells, armed with dynamic repertoire of activating and inhibitory receptors on their cell membranes, engage in a complex interplay upon encountering cancer cells, leading to their activating. Developing imaging techniques for the selective monitoring of NK cell activation, especially targeting specific receptors, poses a significant challenge. In this study, we focused on the STAT1 signaling pathway, known to regulate NK cells' anticancer effects. To selectively monitor STAT1 activation, we chose the interferon gamma activating sequence (GAS) within the promoter region where STAT1 binds in NK cells. The sequence "AGTTTCA-TATTACTCTAAATC" was identified as a STAT1-specific binding GAS-motif, responsive to interferon.

The construction of off/on-type NK cells expressing a reporter protein based on activation status involved the use of the pGF2-NFAT plasmid as a backbone. Four copies of the GAS sequence were inserted via genetic recombination, resulting in the creation of the pGF-GAS lentivirus vector. Subsequently, three distinct packaging vectors—pLP1, pLP2, and VSVG—were employed to transfect 293FT cells. This step was crucial for the production of lentivirus particles, which were then used to infect NK cells. This infection process led to the generation of two distinct NK cell lines: one harboring the empty pGF vector (referred to as empty-NK cells) and the other containing the pGF-GAS vector (referred to as GAS-NK cells, as shown in Fig. 1a). Following the transduction, a two-week puromycin selection phase ensured the successful integration of the GAS sequence into the genomic DNA of the NK cells, a fact that was later confirmed through PCR analysis (as detailed in Fig. S1a). Additionally, to ascertain any potential alterations in the cytotoxic properties of these engineered cell lines, comparative experiments were conducted using K562 cells. The results indicated no significant differences in cytotoxicity between the empty-NK and GAS-NK cell lines, as demonstrated in Fig. S1b. Additionally, experiments to compare the cancer cell cytotoxicity of empty-NK and GAS-NK cells under IFNα treatment conditions. there was no significant difference in the cancer cell cytotoxicity between empty-NK and GAS-NK cells in the presence of IFNα. This comparison demonstrates that while IFNα enhances the cytotoxic activity of both empty-NK and GAS-NK cells, the engineered GAS-NK cells do not exhibit superior cytotoxicity compared to empty-NK cells under these conditions. Therefore, the benefit of GAS-NK cells appears to be STAT1 dependent but not significantly different from empty-NK cells when both are treated with IFNα.

Stimulation of GAS-NK cells with interferon alpha (IFNα; 5000 IU) demonstrated increased luciferase activity (Fig. S1d), with peak expression observed at 24 hours, showing an increase of approximately 70-folds, as well as elevated GFP expression (Fig. 1b, c). Cellular fluorescence imaging and bioluminescence imaging (BLI) further confirmed the functionality of the GAS-NK reporter, with increased BLI values (photons/sec/cell)

proportional to the number of GAS-NK cells treated with IFNα (Fig. S1e, f). Concurrent CD45 and GFP expression was also observed in cell imaging (Fig. S1e).

To examine the correlation between luciferase activity and IFNγ expression, we treated GAS-NK cells with IFNα at different time points and measured IFNγ mRNA levels. An increase was observed at 24 hours, sustained up to 72 hours (Fig. S1g). While luciferase activity peaked at 24 hours and began to decrease after 48 hours, IFNγ levels remained elevated until 72 hours (Fig. S1g), indicating a difference in dynamics. The difference can be attributed to the nature of luciferase as an artificial reporter protein with different expression and degradation kinetics compared to the endogenous cytokine IFNγ. The luciferase assay reflects immediate transcriptional activity that peaks and declines faster, whereas IFNγ expression is regulated to maintain prolonged mRNA and protein levels, indicating sustained immune activation. The shorter half-life of luciferase further explains the quicker decrease in its activity, while IFNγ levels continue to be stable.

An immunoblot assay was conducted to directly confirm the activation of STAT1 in GAS-NK cells following IFNα treatment. Upon treatment with IFNα for 24 hours, a notable increase in STAT1 protein expression was observed. Furthermore, phosphorylation at S701 and Y727, which are key phosphorylation sites of STAT1, was also observed (Figs. 1d and S1j). Complementing this, chromatin immunoprecipitation (ChIP) assays provided direct evidence of STAT1 activation in GAS-NK cells, with increased protein expression and phosphorylation at key sites within the promoter region (Fig. S1h). These results collectively demonstrate that STAT1 protein in GAS-NK cells is capable of binding to the GAS site, leading to an enhanced expression of the reporter protein in response to stimulation.

Additionally, to assess real-time in vivo biodistribution monitoring of GAS-NK cells, the cells were injected into the right flank of nude mice, followed by intraperitoneal IFNα injection and subsequent BLI imaging after 3 hours (Fig. S1i). Pre-activated GAS-NK cells, prepared by 16-hour pre-treatment with IFNα, exhibited an immediate luminescent signal post-intravenous injection into SCID mice, indicating feasible real-time monitoring in vivo (Fig. 1e). These results validated that GAS-NK cell activation can be monitored effectively at both cellular and systemic levels.

### Enhanced cytokine response and STAT1 activation in GAS-NK cells following IFNα stimulation

Upon activation, NK cells are known for their robust production of various cytokines and chemokines, such as IFNγ, TNFα, GM-CSF, CCL1-5, and CXCL8. These molecules play a pivotal role in stimulating other innate and adaptive immune cells, as well as modulating their functions. In our study, we explored the impact of STAT1 activation signaling in GAS-NK cells, focusing on both the enhancement of reporter protein expression and changes in extracellular cytokine release, alongside the associated intracellular signaling alterations.

Cytokine array analysis revealed increased production of IFNγ, FasL, IL-10, IL-1ra, and CTLA8 in IFNα-treated GAS-NK cells, in comparison to the control group (Figs. 2a and S2a). This increase in cytokine production upon IFNα treatment not only underscores the diverse roles of NK cells in immune regulation but also highlights the potential therapeutic implications of modulating NK cell activity.

To further elucidate transcriptomic alterations, we performed bulk RNA sequencing on 24-hour post-IFNα treatment, which showed peak reporter gene expression, as well as on untreated control samples. This approach revealed a total of 440 differentially expressed genes (DEGs): 287 genes with increased expression and 153 genes with decreased expression, all characterized by more than a 2-fold change with p-values less than 0.05 (Fig. 2b). Heatmap analysis distinctly separated the expression patterns of the untreated and treated samples, as shown in Supplementary Fig. 2b. Subsequent ontology analysis, referencing BioCarta 2016 pathways, predicted significant involvement of immune-related signals, including the IFNα, IL1R, IL10, and IFNγ pathways (Fig. 2c). This was further supported by Gene Set Enrichment Analysis (GSEA), which verified the activation of

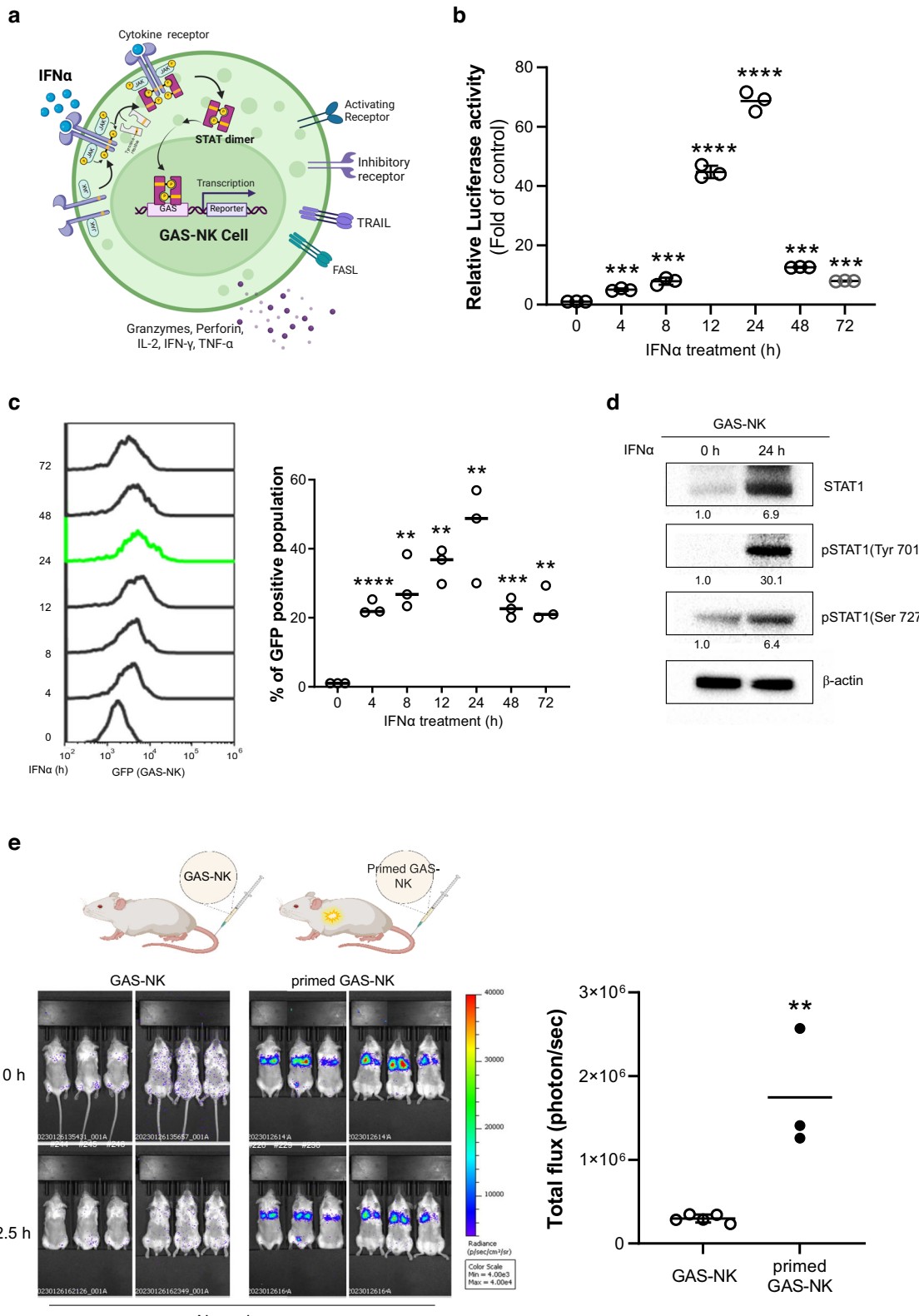

**Fig. 1 | Visualization of IFNα-activated NK cell monitoring in vitro and in vivo.** **a** Illustrative diagram depicting the STAT1 activation mechanism in GAS-NK cells (created with BioRender.com). **b** Graph representing relative luciferase activity in GAS-NK cells over different treatment durations with IFNα (5000 units/mL). ($n$ = 3, samples for all time points). **c** Quantitative analysis highlighting the increase in GFP-positive GAS-NK cell populations in response to IFNα exposure. ($n$ = 3). **d** Immunoblot assay results illustrating the phosphorylation and total protein expression levels of STAT1 in GAS-NK cells, comparing IFNα-treated and untreated conditions. **e** Comparative in vivo imaging of control versus IFNα-primed GAS-NK cells ($5 \times 10^6$/mouse), demonstrating significant differences in luminescent signaling. ($n$ = 3 for both GAS-NK and primed GAS-NK). Statistical significance between groups was determined using paired Student's $t$-tests, with significance levels denoted as follows: **$P < 0.01$; ***$P < 0.001$; ****$P < 0.0001$; ns, not significant.

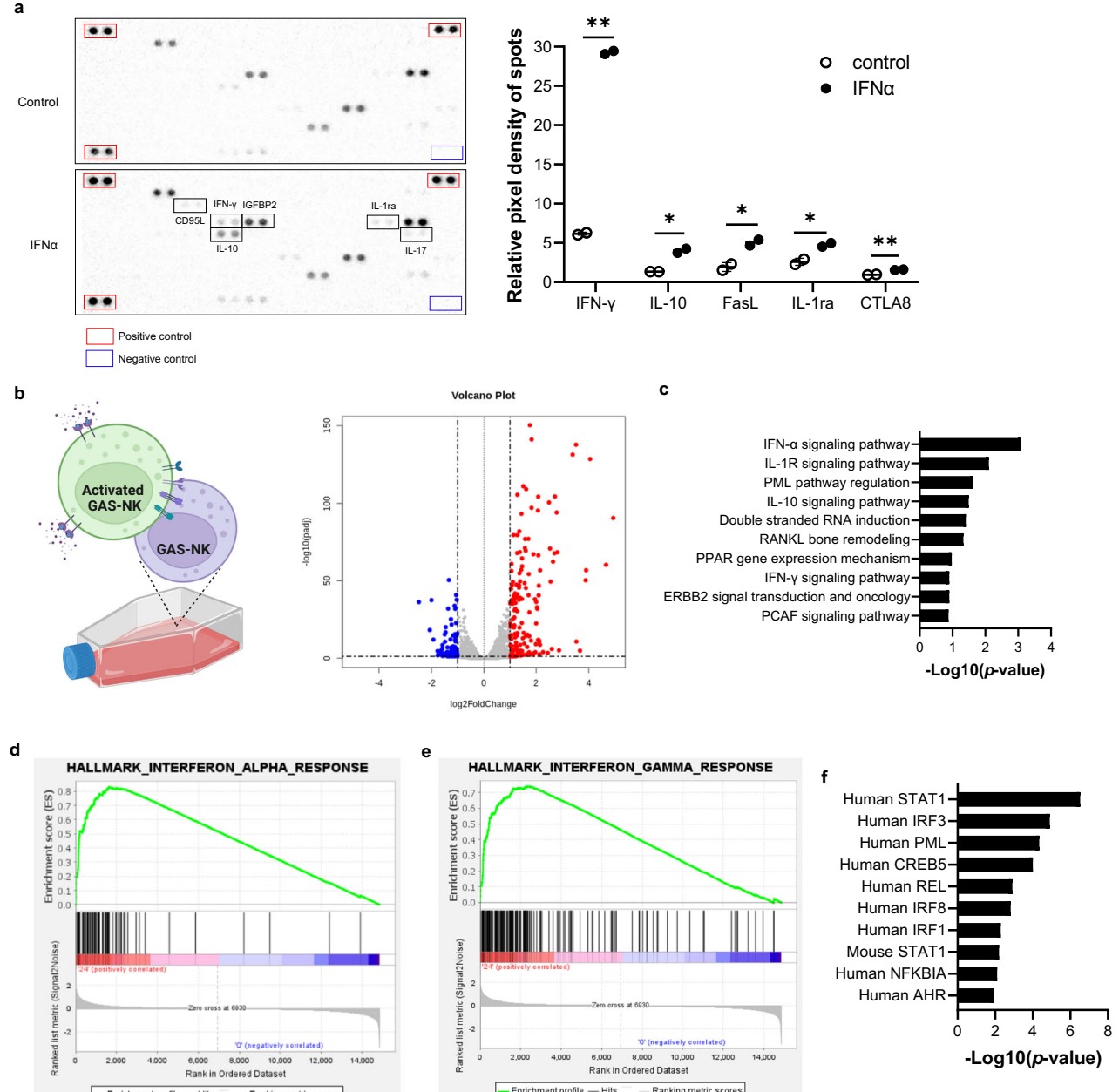

**Fig. 2 | Comparative analysis of GAS-NK cells with and without IFNα treatment.** **a** Human cytokine array analysis contrasting the secretion profiles of various cytokines by GAS-NK cells treated with IFNα for 24 hours versus untreated cells. The array images depict cytokine profiles for both conditions, with the upper image showing untreated GAS-NK cells and the lower one showing IFNα-treated cells. The accompanying bar graph quantifies cytokine levels, highlighting differences in the secretion of cytokines such as IFNγ, IL-10, Fas ligand, IL-1ra, and CTLA8 between the groups. ($n$ = 2; significance levels are denoted as follows: *$P < 0.05$; **$P < 0.01$). **b** A volcano plot highlights differentially expressed genes in GAS-NK cells following 24-hour IFNα treatment compared to untreated cells, with significant upregulation (red dots) and downregulation (blue dots) indicated. **c** A clustered bar chart presents Gene Ontology (GO) analysis of differentially expressed genes (DEGs), focusing on various signaling pathways. GSEA enrichment plots display gene clusters enriched in IFNα-treated GAS-NK cells over 24 hours, emphasizing (**d**) response to IFNα and (**e**) response to IFNγ. **f** Clustered bar chart analysis identifies transcription factors, with human STAT1 showing the highest significance.

signaling mechanisms related to both IFNα and IFNγ responses following IFNα treatment, aligning with the cytokine array findings (Fig. 2d, e). A comprehensive list of signaling genes involved in these responses is presented in Supplementary Fig. 2c, d.

Additionally, TRUST transcriptional regulator prediction analysis identified human STAT1 as a highly significant transcriptional regulator in activated GAS-NK cells (Fig. 3f). Alongside human STAT1, other key transcriptional regulators such as IRF3, PML, CREB5, REL, IRF8, NFKBIA, and AHR were also predicted to be involved, all of which are crucial in modulating intracellular immune responses through interactions with STAT1. Further transcriptome analysis focusing on other STAT family members showed that STAT1 had the highest basal level expression and exhibited an approximate 4-fold increase compared to the control group (Fig. S2e). Although STAT2 expression increased about 5-fold upon IFNα treatment, its lower basal expression suggested a more prominent role for the STAT1 signaling pathway in the activation of GAS-NK cells following IFNα treatment. This indicates that STAT1 activation in GAS-NK cells influences both the expression of internal signaling proteins and the secretion of key cytokines, potentially impacting broader immune responses.

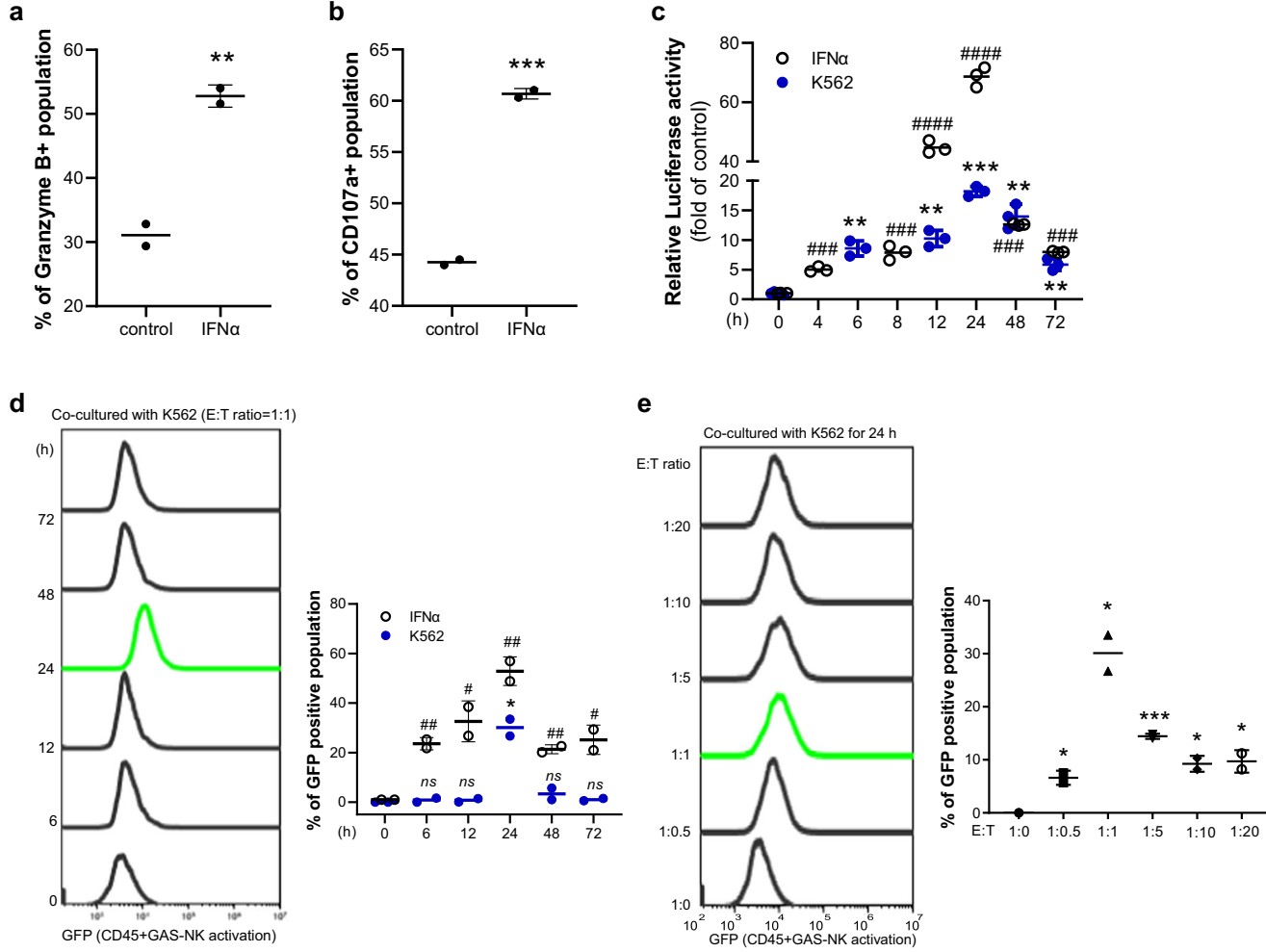

**Fig. 3 | Assessment of enhanced cytotoxicity and selective imaging in IFNα-stimulated GAS-NK cells co-cultured with K562 cells.** Increased expression of cytotoxic markers granzyme B (**a**) and CD107a (**b**) in IFNα-treated GAS-NK cells (5,000 U/mL) compared to controls ($n = 2$, control and IFNα). **c, d** Time-course analysis of luciferase activity ($n = 3$; samples for all time points) and GFP levels ($n = 2$; control and IFNα) in IFNα-treated and K562 co-cultured GAS-NK cells, showing significant variations (IFNα treatment: #; K562 co-culture: *). **e** Quantification of GFP-positive GAS-NK cells at different E:T ratios (1:0, 1:0.5, 1:1, 1:5, 1:10, 1:20) in co-culture with K562 cells ($n = 2$, control and IFNα). Statistical significance between groups was determined using paired Student's $t$-tests, with significance levels denoted as follows: ** or ##, $P < 0.01$; *** or ###, $P < 0.001$; **** or ####, $P < 0.0001$; ns, not significant.

## Enhanced cytotoxic markers in IFNα-stimulated GAS-NK cells and selective imaging during co-culture

Following IFNα stimulation and STAT1 activation, we assessed the expression of cytotoxic markers, granzyme B and CD107a, in GFP-positive GAS-NK cells. Flow cytometry analysis demonstrated significantly elevated levels of both markers in IFNα-treated GAS-NK cells (Fig. 3a, b). To ascertain whether the GFP-positive GAS-NK cells (indicative of reporter signal 'on') were indeed exhibiting anticancer activity, further verification was carried out. This involved gating for STAT1 activation-mediated GFP-positive and GFP-negative GAS-NK cells. Analysis of both GFP-negative and GFP-positive cells in the same IFNα-treated GAS-NK cell sample (Fig. S3a) revealed that the expressions of IFNγ, granzyme B, and CD107a were markedly higher in the GFP-positive cells compared to the GFP-negative cells (Fig. S3b–d). Consequently, this finding confirms that key markers of anticancer activity are predominantly expressed in GFP-positive (signal 'on') GAS-NK cells. This suggests that within a heterogeneous cell population, activated GAS-NK cells can be selectively identified and labeled, providing crucial support for the potential of these cells in experimental applications. This selective labeling paves the way for targeted investigations within the same experimental cell group.

To explore the potential of utilizing STAT1-mediated activated GAS-NK cells, we investigated whether activation could be induced through co-culture with cancer cells. Focusing on their cancer cell-killing efficacy, we co-cultured GAS-NK cells (effector) with K562 blood cancer cells (target) at a standard effector:target (E:T) ratio of 1:1. Within 24 hours of co-culture, both the reporter luciferase activity and GFP intensity in the GAS-NK cells peaked (Fig. 3c, d). The reporter activity at this time point was found to be approximately one-third of the level observed with IFNα treatment, indicating that GAS-NK cell activation via co-culture is detectable but less pronounced compared to direct IFNα stimulation (compared with Figs. 1b and 3c).

To further refine the experimental conditions, we maintained the co-culture condition for 24 hours, identified as the time of highest activity, and then explored a range of E:T ratios (1:0, 1:0.5, 1:1, 1:5, 1:10, and 1:20). This analysis revealed that the maximal GFP activity in GAS-NK cells was achieved when co-cultured with K562 cells at an E:T ratio of 1:1 (Fig. 3e), suggesting this ratio to be the most effective for eliciting GAS-NK cell activation.

## GAS-NK cell activation in co-culture with solid tumor cells and its implications for STAT1 signaling

To explore the feasibility of monitoring GAS-NK cell activation during co-culturing with solid tumor cells, we examined various cancer cell lines. The GAS-NK cells with the A549 human lung cancer cell line for varying

durations (0, 6, 12, 24, 48, 72 h). In each instance, we evaluated the GFP-positive population of GAS-NK reporters and luciferase activity. Similar to the co-culture with K562 cells, the highest levels of reporter luciferase activity and GFP intensity in the A549 cells were observed at 24 hours, showing an approximately 6-fold and 5-fold increases, respectively (Fig. 4a, b). While the GFP-positive population in the K562 cell co-culture was analyzed only at the 24-hour mark, in the A549 cell co-culture, an increase was noted up to 24 hours, albeit with a somewhat lower rate of increase in luciferase activity compared to the K562 cell co-culture.

Subsequently, after establishing 24 hours as the time of peak activity, we conducted further verification with various E:T ratios (1:0, 1:1, 1:5, 1:10, and 1:20). Our results indicated that the highest GFP-positive population and luciferase increase occurred when A549 cells were co-cultured with GAS-NK cells at a 1:1 E:T ratio (Fig. 4a, b). We also performed experiments to determine if such activation of GAS-NK cells would occur when co-cultured with normal lung cells (WI-38, MRC5). Under all conditions, no increase in GFP-positive GAS-NK cell activation was observed when co-cultured with normal lung cells at any time or E:T ratio (Fig. S4c–f).

To further assess whether GAS-NK cell activation could be monitored in tumor tissue in vivo, additional experiments were conducted using various solid tumor cell lines co-cultured in the form of 3D spheroids. We observed luciferase activity in GAS-NK cells co-cultured with representative organ solid tumor cell lines (breast cancer, MDA-MB-231; stomach cancer, HCT116; liver cancer, HepG2; lung cancer, A549). Compared to GAS-NK cells alone, co-culture spheroids with MDA-MB-231 resulted in higher increase in luciferase activity in GAS-NK cells, with lower induction observed in A549 spheroids (Fig. 4c).

Given these differences, we hypothesized that variations in STAT1 activation in GAS-NK cells may be linked to the different cytokines secreted by each cancer cell. We assumed that within a tumor microenvironment, diverse stimulatory signals induce an immune response, and that even under in vitro conditions, each cancer cell line would secrete a distinct quantity and type of cytokines. Thus, we cultured four cancer cell lines (each with 50,000 cells in a 96-well plate) and collected the culture medium after 24 hours to analyze cytokine production (Fig. S5a). A total of 105 cytokines were assessed using a cytokine array (Fig. S5b). Among these 105 cytokines, 16 known to directly or indirectly activate STAT1 were identified using Ingenuity pathway analysis (IPA), a knowledge-based signal network analysis tool (Fig. S5c, d). Using Image J Fiji software, we quantified the pixel intensity of cytokine spots secreted by each cancer cell line and visualized the data as a tile plot. The tile plot illustrated the relative distribution of cytokine production, categorizing levels into ranges from 0 to 120. MDA-MB-231 cells, which exhibited the highest increase in GAS-NK luciferase activity, also had the highest levels of analyzed cytokines, while A549 cells, showing the lowest NK activity, produced the fewest total cytokines (Fig. 4d). Using IPA, a knowledge-based signal network analysis tool. It was predicted that 16 cytokines, which are expressed at the highest levels in four solid tumor cell lines, can activate STAT1 directly or indirectly (Fig. S5c). Additionally, it was computed the z-score of pathway analysis on "Binding of STAT binding site" based on increased cytokine production in each solid cancer cell line, confirmed activation specifically in the MDA-MB-231 (Fig. S5d). Considering these cytokine array results, the differences in GAS-NK activation levels among co-cultured cancer cells may stem from variations in the production of cytokines related to STAT1 signaling activation.

## Differential activation of GAS-NK cells in response to tumor microenvironmental cytokines

Numerous studies have highlighted the inhibition of NK cell activity by transforming growth factor-β (TGFβ) cytokines within the tumor microenvironment (TME). To evaluate the resilience of GAS-NK cells in such conditions, we treated these cells with TGFβ and monitored reporter activity. There are three types of TGFβ receptors: TGFβR1, TGFβR2, and TGFβR3. TGFβR1 is recognized as the central receptor in TGFβ signaling, playing a critical role across various cell types. TGFβR2 associates with TGFβR1 to

facilitate TGFβ signaling, while TGFβR3 primarily mediates the binding of TGFβ without directly participating in signaling. Typically, TGFβR1 shows the highest expression levels, followed by TGFβR2 and then TGFβR3. Our RNA sequencing results confirmed that all three receptors are expressed in GAS-NK cells, the expression pattern of TGFβ receptors in GAS-NK cells was found to align with established scientific literature (Fig. S6b).

Based on our observations, the GFP(+) population indicating activation decreased by approximately 50% after exposure to two concentrations of TGFβ, in comparison to the negative control group. Furthermore, although there was an increase in the GFP(+) population following treatment with IFNα, exposure to two concentrations of TGFβ resulted in a subsequent decrease of approximately 10% (Fig. 5a). This finding was further supported by fluorescence imaging, which demonstrated a reduction in strong fluorescence signal post-IFNα treatment upon the addition of recombinant TGFβ (Fig. S6a). Furthermore, the luciferase(+) level in GAS-NK cells, which was elevated following IFNα treatment, exhibited a decrease of approximately 40% after exposure to two concentrations of TGFβ (Fig. 5b).

Considering the influence of the tumor microenvironment on cytokine production, we explored whether GAS-NK cell activity varied with different levels of IFNα cytokine. Figure 4c and d showed that GAS-NK cell activity was lowest when co-cultured with NSCLC A549 cells, which secreted lower amounts of cytokines compared to other cancer cell lines. To further investigate this, we prepared an A549-IFNα cell line using a lentivirus system and quantified IFNα production. In a 2D culture of 5000 A549 cells over 72 hours, IFNα levels reached 0.015 ng, whereas A549-IFNα cells produced significantly higher levels of IFNα at 15.53 ng/ml—a thousand-fold increase compared to the standard A549 2D culture (Fig. 5c). Moreover, in 3D cultures, IFNα production was even higher (Fig. 5c). To assess whether cytokines in the tumor microenvironment could induce GAS reporter activity, GAS-NK cells were treated with culture medium sampled from 3D cultures of A549-IFNα cells, confirmed to secrete IFNα. Treatment with 10 μl of this medium (approximately 26 ng/ml IFNα) resulted in more than a two-fold increase in GAS luciferase activity. For comparison, we used 5,000 IU of IFNα (about 34 ng/ml) as a positive control (Fig. S6c). Furthermore, when GAS-NK cells were co-cultured with A549 and A549-IFNα cancer cells at an E:T ratio of 1:1, the GFP-positive population increased over time, confirming reporter activation by IFNα (Fig. 5d). Additionally, in experiments involving 3D spheroids co-cultured with GAS-NK cells, we detected an increase in luciferase activity and GFP fluorescence intensity in groups co-cultured with IFNα-producing A549-IFNα cells (Figs. 5e and S6e, S6f). These findings confirm that GAS-NK cell activation is induced by IFNα present in the tumor microenvironment, establishing a quantitative correlation between the level of IFNα and the activation of GAS-NK cells.

## GAS-NK cell targeting efficacy in lung metastasis models through selective imaging

Our study aimed to overcome current limitations in monitoring NK cell therapy, where imaging typically involves the entire transplanted cell population, despite only a fraction being activated and capable of targeting cancer cells. We sought to explore the feasibility of selectively imaging NK cells activated via GAS signaling, as these cells are likely to possess cancer-killing abilities.

Initially, we conducted a pilot experiment by injecting A549 cancer cells ($2 \times 10^6$ cells/mouse) into the tail vein of SCID mice to create a lung metastasis model. On day 21 after cancer cell administration, GAS-NK cells ($5 \times 10^6$ cells/mouse) were injected via the tail vein into A549 lung metastasis SCID mice, and bioluminescent imaging (BLI) was monitored from 0 to 48 hours post-injection (Fig. S8a). The data showed higher BLI values at 3 hours, indicating that the in vivo environment indeed differs from in vitro conditions. Despite these differences, it was confirmed that monitoring GAS-NK cell activation in vivo is feasible. On day 28 post-cancer cell administration, one mouse from each group (normal vs. lung metastasis) was sacrificed to verify the model. Lungs in the A549 lung metastasis model exhibited enlargement, irregular surface texture, and observable white

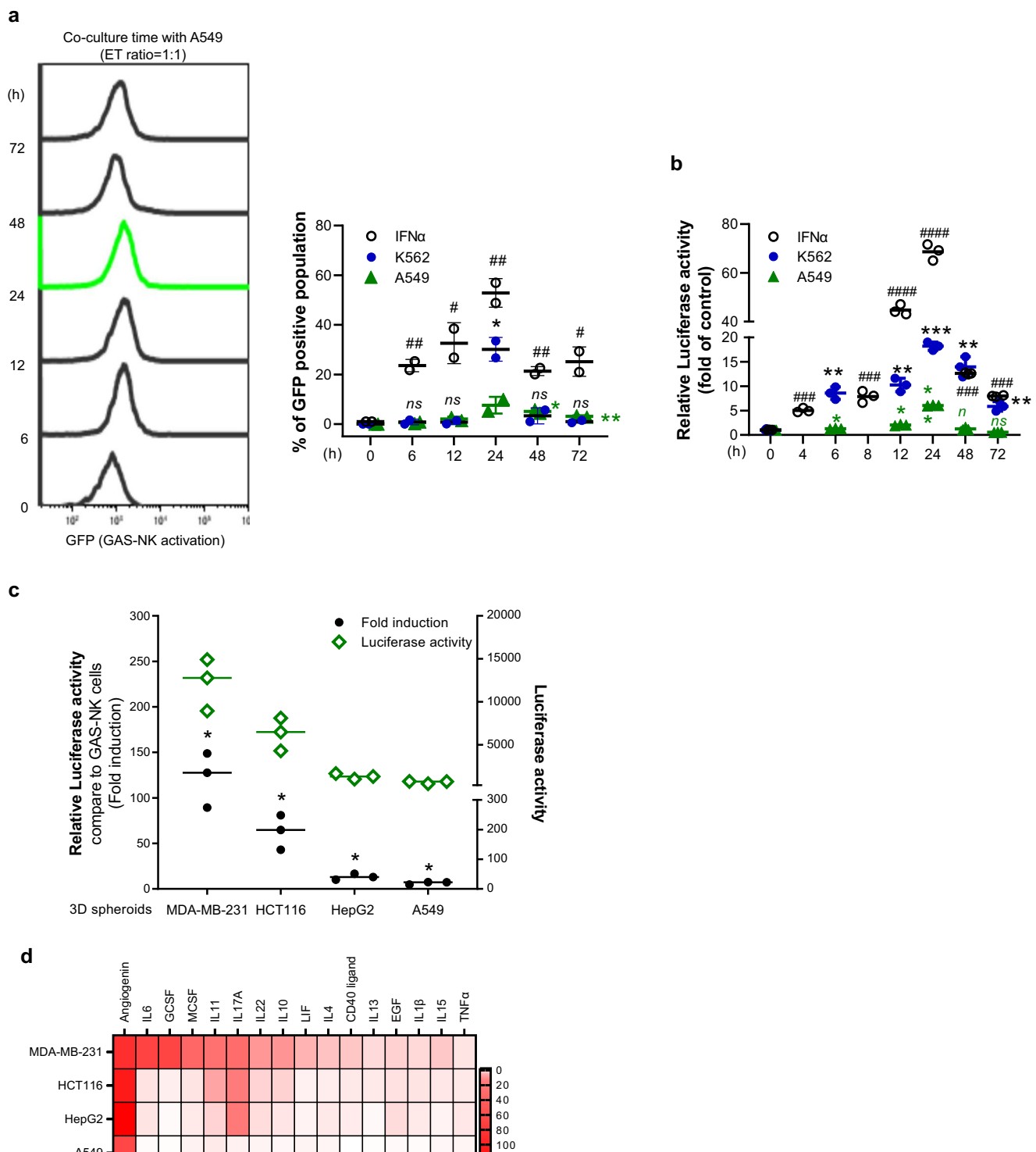

**Fig. 4 | Impact of GAS-NK cell activation in co-culture with solid cancer cells.**
**a** Histogram illustrating GFP intensity changes over time in GAS-NK cells co-cultured with A549 cells at an E:T ratio of 1:1. The accompanying quantitative graph combines these results with those from IFNα treatment and K562 cell co-culture (Fig. 3d), highlighting significant variations (A549 co-culture: *; IFNα treatment: #; K562 co-culture: *, blue color). ($n = 2$, samples for all time points). **b** Luciferase activity in GAS-NK cells co-cultured with A549 at an E:T ratio of 1:1, compared with results from IFNα treatment and K562 cell co-culture (Fig. 3c) (A549 co-culture: *; IFNα treatment: #; K562 co-culture: *; blue color). ($n = 3$, samples for all time points). **c** Line graph comparing luciferase activity of GAS-NK cells co-cultured with various solid tumor cell 3D spheroids (MDA-MB-231, HCT116, HepG2, A549) to GAS-NK cells alone. ($n = 3$, all other samples). **d** Tile plot indicating cytokine production levels in culture supernatants from four solid cancer cell lines, with color intensity changing from white to red indicating higher cytokine levels. Statistical significance between groups was determined using paired Student's $t$-tests, with significance levels denoted as follows: * or #, $P < 0.05$; ** or ##, $P < 0.01$; *** or ###, $P < 0.001$; **** or ####, $P < 0.0001$; ns, not significant.

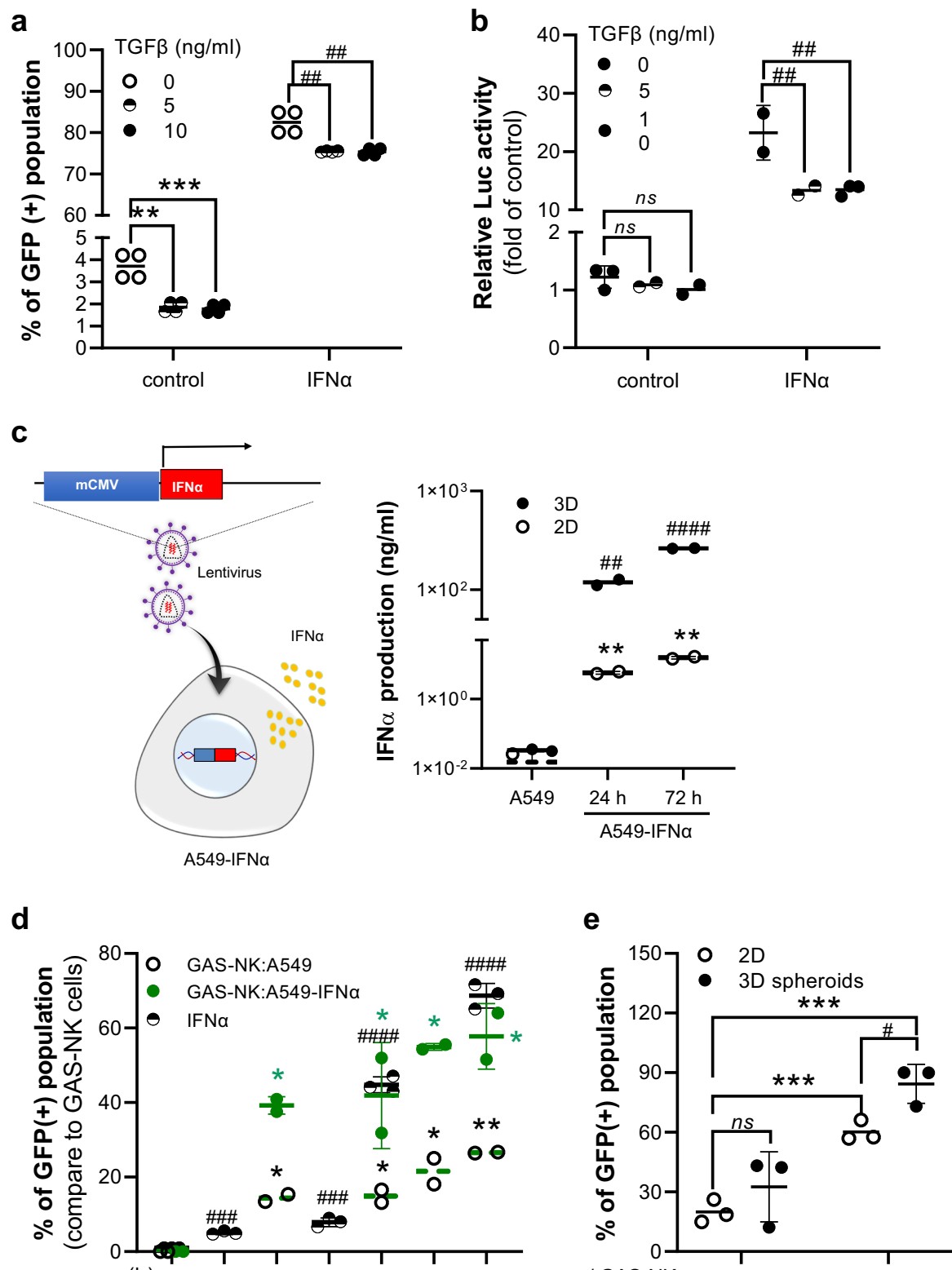

**Fig. 5 | Analyzing GAS-NK cell response to tumor microenvironment cytokines.**
**a** Response of GAS-NK cells to TGFβ (5 or 10 ng/ml) treatment, illustrated by changes in GFP intensity. A quantitative graph compares responses to IFNα and TGFβ treatments. Statistical significance is indicated in comparison to control and IFNα-treated cells (control: *; IFNα treatment: #). ($n = 2$, all other samples).
**b** Luciferase activity of GAS-NK cells under the same treatment conditions, with statistical analysis against control and IFNα treatments (control: *; IFNα treatment: #; $n = 3$, control and IFNα). **c** Schematic of the A549-IFNα cell line and a

comparative analysis of IFNα secretion levels in 2D and 3D cultures of A549 and A549-IFNα cells (statistical significance indicated for 2D vs. 3D comparisons; $n = 3$, control; * and IFNα; #). **d** Percentage of GFP-positive GAS-NK cells over time when co-cultured with A549 or A549-IFNα cells (E:T ratio = 1:1), integrating results from previous figures (IFNα treatment: *; A549: #; A549-IFNα: *, green color; $n = 2$ or 3, samples for all time points). **e** Graph quantifying the GFP( + ) GAS-NK cell population when co-cultured with 2D or 3D spheroids of A549 and A549-IFNα cells for 24 hours ($n = 3$, samples for all time points). (****P* < 0.001; ns, not significant).

blisters, distinguishing from the normal mice. Once tumor formation was confirmed, GAS-NK cells ($5 \times 10^6$ cells/mouse) were administered via the tail vein to both groups. BLI imaging, conducted 10 minutes post-intraperitoneal D-luciferin injection, revealed significantly higher activation of GAS-NK cells in the lung area of A549 lung metastasis model mice compared to normal mice (Figs. 6c and S7b). Lung metastatic tumor formation was further validated through H&E staining and E-cadherin IHC (Figs. S7a, 7c), with H&E staining indicating typical metastatic tumor characteristics.

In our study, we also explored the targeting ability of GAS-NK cells in a lung cancer metastasis model induced in mice using A549 cells. Specifically, we examined whether GAS-NK cells, administered intravenously, were able to home specifically to tumor sites within the lung tissue. The tissue samples were processed for histological examination using H&E staining and were placed on a thermoplastic film for precise dissection. Using laser capture microdissection, we successfully isolated distinct tissue regions containing either normal cells or cancer cells adjacent to the tumor. The specificity of GAS-NK cell homing to the tumor sites was confirmed through the gene expression analysis of GAS, CD56, and NKG2D markers.

Additionally, we performed gene expression analysis for pro-inflammatory cytokines TNF-α and IL-6 in these isolated tissue samples. Our results indicated no significant change in TNF-α expression between adjacent normal and tumor tissues. However, IL-6 expression was notably increased in the tumor tissues, suggesting an inflammatory response potentially contributing to the anti-tumor activity of GAS-NK cells (Fig. S7d). This approach allowed us to effectively determine the localization and activity of GAS-NK cells in the tumor microenvironment, providing valuable insights into their targeting efficiency and potential therapeutic application in lung cancer metastasis. Further details and visual evidence of these findings can be seen in Fig. S7d.

To validate the activity of GAS-NK cells in tumors, a repeated experiment was conducted over 21 days, increasing the number of subjects per group for added experimental reliability (Fig. 6a). The formation of the A549 lung metastasis model ($5 \times 10^6$ cells/mouse) was monitored using MRI imaging to track metastatic tumor burden (Fig. 6b). GAS-NK cells were injected into the tail vein of mice 20 days post-tumor inoculation, a time point confirmed for tumor formation via MRI. BLI imaging, conducted 10 minutes after intraperitoneal D-luciferin injection, again demonstrated significantly higher activation of GAS-NK cells in the lung area of A549 lung metastasis model mice compared to normal mice (Fig. 6c). On day 21, all mice were sacrificed, and lungs were collected for further analysis. Half of the lung tissue underwent H&E staining, while cells from the other half were isolated for human NK cell marker analysis, focusing on CD56 and CD107a. H&E staining results mirrored those of the pilot experiment, with observable tumor characteristics (Fig. 6d). Interestingly, the analysis of NK cell markers showed no significant difference in the number of NK cells present in the lung tissues of both models. However, CD107a, an indicator of functional activity, was highly expressed in the GAS-tumor model (Fig. 6e). These in vivo findings indicate that GAS-NK cells can not only home to tumor sites but can also be specifically imaged to highlight cells exhibiting heightened antitumor activity.

In the tumor metastasis model, we observed a trend towards reduced tumor size with NK cell treatment compared to PBS. However, when comparing the anti-cancer effects between empty-NK cells (negative control) and GAS-NK cells, no significant differences in overall survival were observed (Fig. S8b). This indicates that while both empty-NK and GAS-NK cells were effective in reducing tumor size, the engineered GAS-NK cells did not provide a significant improvement in overall survival compared to empty-NK cells. These findings suggest that while GAS-NK cells show potential in targeting tumors, further optimization and investigation are needed to enhance their overall therapeutic efficacy.

### Selective activation of GAS-NK cells in cancer organoids and integration into CAR-NK cells

In our conclusive analysis, we sought to evaluate the clinical applicability of GAS-NK cells by examining their activation in co-culture with patient-derived tumor organoids. Initially, we cultured human iPSC-derived liver and intestine organoids, observing no significant changes in the reporter signal when these normal organoids were co-cultured with GAS-NK cells, as indicated by consistent luciferase activity levels (Fig. S9a). Subsequent fluorescence imaging was performed with three different types of tumor organoids co-cultured with GAS-NK cells. Notably, the GFP intensity in GAS-NK cells was notably high when co-cultured with colon adenocarcinoma organoids (SNU-977) (Fig. 7a). In contrast, while GFP intensity increased in co-cultures with gastric adenocarcinoma (SNU-1) and cellular adenoma (SNU-1181) organoids, the levels were comparatively lower than those observed with SNU-977 (Fig. S9b). Additionally, in co-cultures with SNU-977 organoids, we observed a relatively high expression of CD107a, a key activation marker of NK cells, suggesting a differential response of GAS-NK cells to various organoid types (Fig. 7b). While further validation is necessary to determine if GAS-NK cell reporter activity can predict responders and non-responders to NK cell therapy, our results indicate that GAS-NK cell activation varies depending on the patient or cancer type.

In light of recent advancements in CAR-NK cells to enhance NK cell anticancer activity, we also explored the integration of the GAS system into αPDL1-CAR-GAS-NK cells. These cells were designed with an intracellular signaling moiety, including CD28, DAP10, and CD3ζ, along with hinge and transmembrane regions[14]. To assess any functional impact of the GAS reporter system on PDL1-CAR-NK cells, we evaluated their cytotoxicity in co-cultures with AU565 cells (low PDL1 expression) and H1975 cells (high PDL1 expression). No notable change in anticancer activity was observed following GAS system integration (Fig. S10b). Additionally, we tested the functionality of the GAS reporter system in αPDL1-CAR-GAS-NK cells, stimulating them with IFNα and subsequently analyzing GFP intensity via flow cytometry. Post-IFNα treatment, αPDL1-CAR-GAS-NK cells displayed a 41.2% increase in GFP intensity (Fig. S10a). Thus, our findings suggest that this GAS-dependent selective monitoring system of NK cell anticancer activity holds potential for application in the development of CAR-NK cell therapies.

## Discussion

Immunotherapies, which exploit immune cell cytotoxicity, hold significant promise for tumor elimination, with approaches involving genetically engineered cells or antibodies targeting immune checkpoint receptors[14–18]. The administration of expanded NK cells following in vitro cultivation has emerged as a notable strategy in immunotherapy, capitalizing NK cells' innate ability to target and eliminate cancer cells. This potent and targeted approach presents a promising method for combating various cancer. In the landscape of NK anti-cancer immune cell therapy, the assessment of NK cell bio-tracking technology conventionally focused on evaluating anti-cancer efficacy and tumor size reduction. However, the traditional method involves monitoring all transplanted cells, despite reports suggesting that only a fraction of these cells are actually effective against cancer[19]. Clinical trials on adoptive NK cell transfer have yielded positive outcomes in diverse cancers, including acute myeloid leukemia and ovarian cancer[20–24]. Nevertheless, instances exist where NK cells fail to eradicate tumors[25]. The integration of detectable, activation-dependent reporter activity in NK cells presents a potential avenue for tailoring optimal NK therapies on an individualized basis. STAT1 monitoring offers distinct advantages in this context. While IFNγ and Granzyme B are general markers of NK cell activation, they are also secreted in response to various non-cancerous stimuli, which can confound monitoring efforts. STAT1, however, is crucial in NK cell development, IFNγ production, and cytotoxic function, particularly in the tumor microenvironment. This specificity allows for more precise tracking of NK cells' antitumor activity, minimizing interference from other immune responses.

This study introduces a sophisticated imaging technique designed to selectively track activated NK cells capable of attacking cancer cells within tumors. Through the genetic modification of GAS-NK cells with the STAT1 promoter binding motif, crucial for NK cell function, and incorporation of a reporter gene, we can precisely monitor STAT1's regulatory role. This approach facilitates real-time monitoring and quantification of NK cell

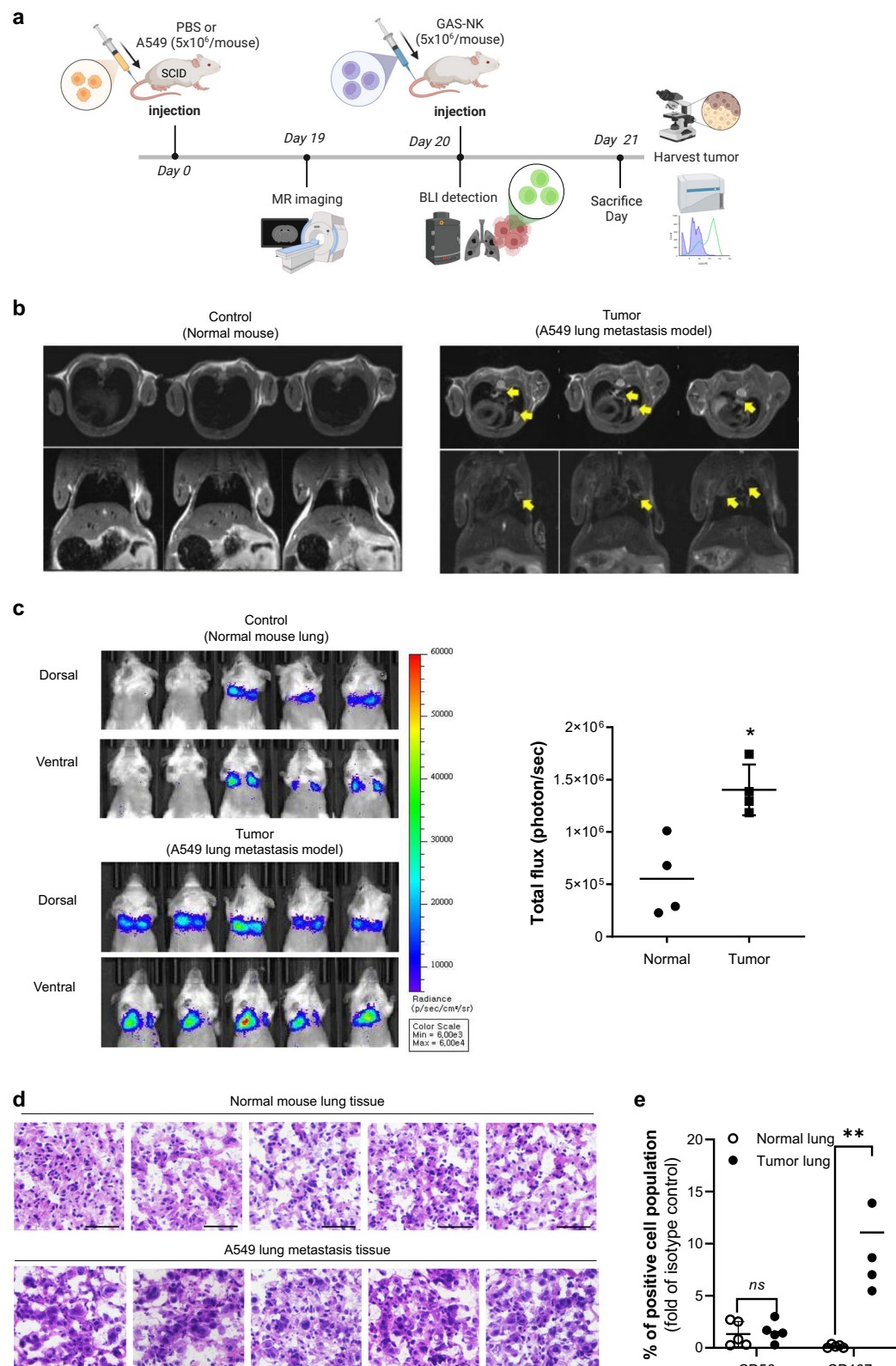

**Fig. 6 | Tumor site-specific activation monitoring of GAS-NK cells in an A549 lung metastasis model. a** Time-lapse schematic of the experimental procedure involving A549 and GAS-NK cell injections, MRI imaging, BLI detection, and tumor tissue analysis in SCID mice (created with BioRender.com). **b** MRI images confirming lung metastasis model formation 19 days post-A549 tail vein injection, with tumors indicated by yellow arrows. **c** BLI images of control (top) and tumor groups (bottom) following tail vein injection of GAS-NK cells into normal and A549 lung metastasis SCID mice. Each image represents an independent experiment, with luminescence quantified as total flux photons/sec from targeted body areas. Statistical significance is indicated (*$P < 0.05$). ($n = 5$, control and tumor). **d** H&E stained lung tissue from normal and tumor-bearing mice, 24 hours post-GAS-NK cell injection. **e** Comparison of the percentages of CD56 and CD107a positive NK cell populations in lung tissues from normal and lung metastasis mice, with statistical significance noted (**$P < 0.01$; ns, not significant). ($n = 3$, control and tumor). Statistical significance between groups was determined using paired Student's $t$-tests, with significance levels denoted as follows: *$P < 0.05$; **$P < 0.01$; ns, not significant.

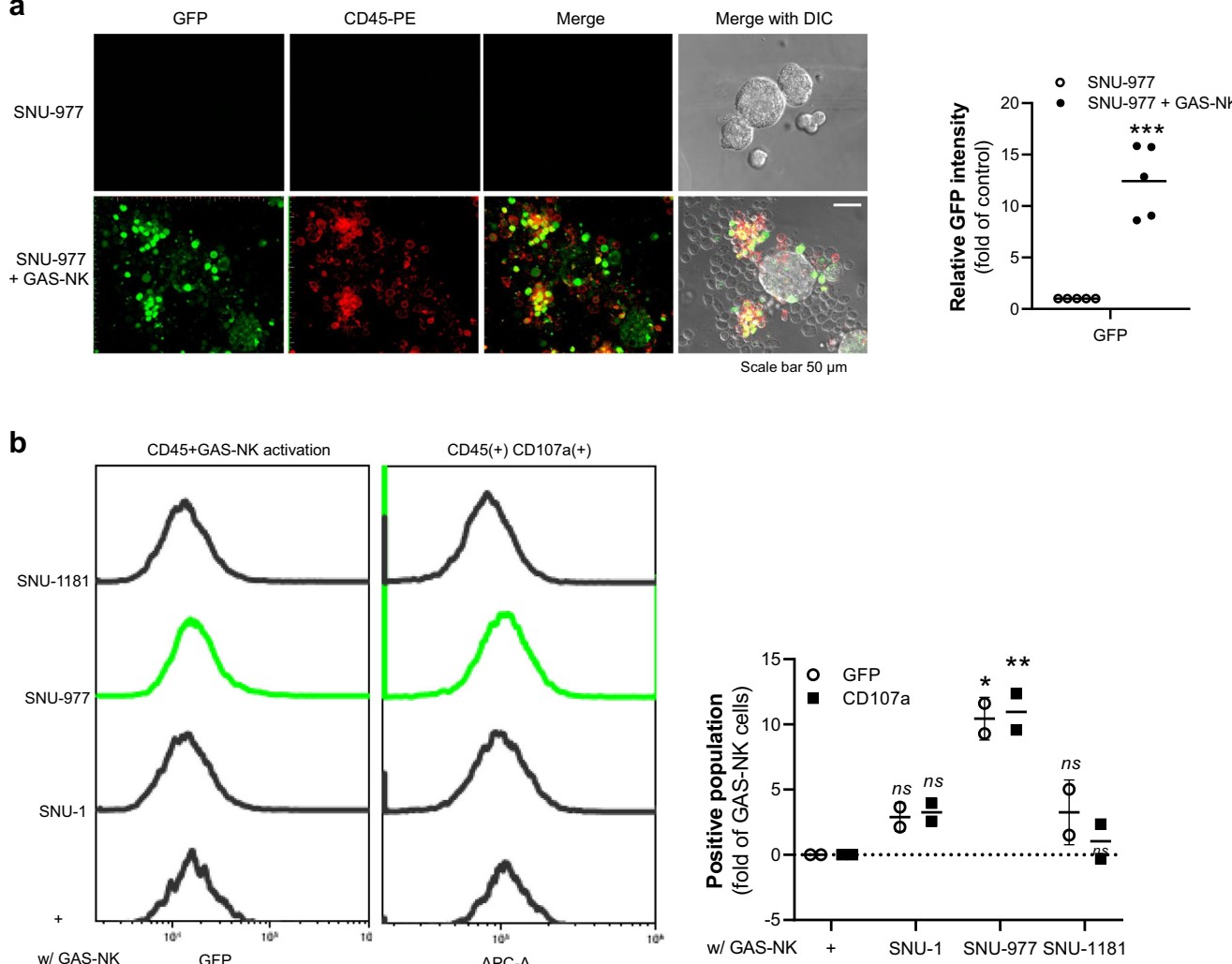

**Fig. 7 | Assessing GAS-NK cell activation with patient-derived cancer organoids.** **a** Fluorescence images of GFP and CD45-PE (red) post co-culture of tumor organoids (SNU-977) with CD45-PE stained GAS-NK cells (left panel). The accompanying graph quantifies GFP intensity in CD45-positive GAS-NK cells (right panel), with statistical significance indicated (**$P < 0.001$). ($n = 5$, SNU-977 and SNU-977 + GAS-NK). **b** Flow cytometry results of GFP and CD107a(+) populations in GAS-NK cells after 24-hour co-culture with three types of cancer organoids (SNU-1, SNU-977, SNU-1181) at an E:T ratio of 1:1. Bar charts depict fold changes in GFP and CD107a(+) populations in CD45( + )-GAS-NK cells ($n = 2$, all other samples). Statistical significance between groups was determined using paired Student's $t$-tests, with significance levels denoted as follows: *$P < 0.05$; **$P < 0.01$; ***$P < 0.001$; ns, not significant.

distribution and activity in vivo. Our results showcase the successful selective monitoring of activated GAS-NK cells, particularly evident in co-cultures with K562 hematological cancer cells and various solid tumors, as indicated by the enhanced activity of GFP and luciferase reporter proteins. We compared the GFP activity of GAS-NK cells at the peak activation time of 24 hours across four groups: GAS-NK alone, IFNα-treated, K562, and A549-IFNα cells. The results indicated that the GFP-positive cells were observed at significantly higher levels in the IFNα-treated group (140.9-fold), followed by K562 (60.3-fold), A549 (15.2-fold), and A549-IFNα cells (105.2-fold) compared to the GAS-NK alone group (Figure S6d). Our findings confirm that IFNα is the most potent stimulator of GAS-NK cell activity, while among the cell lines, A549-IFNα cells showed the highest activation. This comprehensive analysis provides a clearer understanding of the variation and significant differences in GAS-NK cell activation under these conditions.

NK cells release cytolytic granzymes to induce apoptosis in target tumor cells[25–28]. The development of granzyme B-specific substrate-dependent chemiluminescence probes for monitoring NK cell activity has been reported[29]. However, these probes face limitations due to basal-level imaging in NK cells. In contrast, our study's GAS-NK cells utilize STAT1-mediated signaling for reporter expression, offering a more precise and

distinct method for monitoring NK cell activation and function. While this innovative approach allows for the specific detection of NK cells actively engaged in combating tumor cells, highlighting the potential pivotal role GAS-NK cells could play in enhancing the efficacy and precision of NK cell-based cancer therapies, it presents a limitation. Specifically, our method is restricted to monitoring STAT1-mediated NK cell activation and does not facilitate the concurrent measurement of other effector functions, such as those mediated by NFAT. This limitation underscores the need for further research to develop methods capable of simultaneously assessing multiple aspects of NK cell functionality to fully understand their roles in cancer therapy. This study underscores the significant influence of TGFβ on GAS-NK cell responses, indicating the necessity for broader investigations into a variety of factors across different tumor microenvironments.

Moreover, this study highlights the pivotal role that GAS-NK cells play in cancer therapy, demonstrating their proficiency in selective activation, precision imaging, and their ability to integrate seamlessly into sophisticated therapeutic modalities. These accomplishments mark a significant progression in the field of immunotherapy, paving the way for the development of targeted strategies that promise more effective NK cell-based therapies, specifically designed to cater to the individualized needs of patients. By

illuminating the practical therapeutic potential of GAS-NK cells, this research not only contributes to the current knowledge base but also lays the groundwork for future endeavors aimed at enhancing and refining NK cell-based approaches to cancer treatment.

## Materials and methods
### Cell line and culture conditions
Various cell lines were employed in this study, all procured from the American Type Culture Collection (ATCC; Manassas, VA, USA). These included NK-92 (human natural killer cell line), MRC-5 and WI-38 (human normal lung cell lines), A549 (human non-small cell lung cancer cell line), MDA-MB-231 and AU565 (human breast cancer cell lines), HCT116 and HT29 (human colorectal cancer cell lines), and HepG2 (human liver hepatoma cancer cell line). The NK-92 cells were cultured in alpha minimum essential medium supplemented with 12.5% fetal bovine serum (FBS) and 12.5% horse serum, 0.2 mM inositol, 0.1 mM 2-mercaptoethanol, 0.02 mM folic acid, and 200 U/mL recombinant IL-2 (Cat. 200–02, Peprotech, Rocky Hill, NJ, USA). This composition was intended to support the robust proliferation and maintenance of NK-92 cells. The other cell lines were cultured under different conditions tailored to their specific requirements. MDA-MB-231, HCT116, HT29, H1975 and HepG2 cell lines were maintained in Dulbecco's Modified Eagle's Medium (DMEM), whereas MRC-5 and WI-38 cell lines were cultured in Eagle's Minimum Essential Medium (EMEM). The A549 and AU565 cell lines were cultured in RPMI-1640 medium. All these media were supplemented with 10% FBS. The culture environment was kept consistent for all cell lines, maintained in a humidified incubator with 5% $CO_2$ at a temperature of 37 °C, which is critical for optimal cell growth and sustainability. All cell lines were routinely tested for mycoplasma contamination to ensure the integrity of the experiments. Mycoplasma testing was performed using a PCR-based method with the e-Myco PLUS Mycoplasma PCR Detection Kit (Cat. #25237, iNtRON Biotechnology, South Korea), and only mycoplasma-free cell lines were used in this study.

### Cell line production
NK-92 cells with enhanced GAS promoter elements: NK-92 cells were maintained in 24-well plates at a concentration of 500,000 cells per well and transduced with lentiviral particles containing enhanced GAS promoter elements derived from the pGreenFire2-NFAT plasmid (System Biosciences, Cat #, TR451PA-P). Trans-Dux MAX reagent (System Biosciences) was used for efficient transduction. After a two-week puromycin (1 µg/mL) selection process, transduction efficiency was assessed by flow cytometry (CytoFlex, Beckman Coulter), and GFP-positive cells were isolated using an advanced cell sorter (Astrios, Beckman Coulter). Engineered NK-92 cells (pGF-empty and pGF-GAS-NK cells; Empty-NK and GAS-NK cells) contained a quadruple copy of the GAS-responsive sequence (AGTTT-CATATTACTCTAAATC), allowing for heightened responsiveness. A549 cells engineered for IFNα overexpression: A549 cell lines were cultured in 24-well plates at a seeding density of 50,000 cells per well. These cells were then infected with lentiviral particles engineered for IFNα overexpression using the pGEM-IFNα vector (Cat# HG12341-G, Sino Biological Inc, Wayne, PA, USA) and the same transduction reagent as above. A two-week G418 (50 µg/mL) selection process was conducted to ensure the survival of successfully modified cells. This process aimed to establish A549 cell lines with heightened IFNα expression for more robust experimental outcomes.

### 3D spheroids culture conditions
Cancer cells were harvested and seeded at a density of 1,000 cells per well in round-bottom 96-well ultra-low attachment plates (CellCarrier Spheroid ULA 96-well Microplates, PerkinElmer, Waltham, MA, USA). To facilitate spheroid formation, these plates underwent centrifugation at 300 × g for 15 minutes at room temperature. Subsequently, they were incubated at 37 °C with 5% CO2 for 72 hours, and then the culture medium was replaced with fresh medium. Fluorescence images of the spheroids were captured using a ZOE™ Fluorescent Cell Imager (Bio-Rad).

### Cancer organoid culture conditions
Cancer organoids (SNU-1, SNU-977, SNU-1181) were obtained from the Korea Cell Line Bank (KCLB, Seoul, Republic of Korea). Organoids were cultured in a specialized medium composed of 40% w/v DMEM/F12, 50% w/v L-WRN conditioned medium, and supplemented with 1× B27, 50 ng/mL human EGF, 10 ng/mL human FGF-10, 10 mM nicotinamide, 1.25 mM N-acetylcysteine, 500 nM A83-01, 3 µM SB202190, 10 nM prostaglandin E2, and N2 supplement. The culture was maintained in a humidified incubator with 5% CO2 at 37 °C. For co-culture experiments with NK cells, the organoids were first dissociated into single cells and then seeded at a density of 2500 cells per well in ultra-low attachment (ULA) round-bottom 96-well plates. After a 3-day incubation, GAS-NK cells were added at a concentration of 5000 cells per well and co-cultured with the organoids for 24 hours in NK cell-specific medium.

### Microscopy imaging
GAS-NK cells, stained with CD45-PE antibody for 4 hours prior to co-culture, were seeded at 25,000 cells per well in IBIDI chamber slides (Ibidi, Martinsried, Germany). They were incubated for 24 hours at 37 °C at an effector to target (E:T) ratio of 1:5. For the positive control, GAS-NK cells were treated with IFNα. To minimize imaging time and prevent photodamage, images were captured with 1.2–1.3 µm z-slices and a pixel size of 1,024 nm with 4× line averaging. Sequential images were collected using a laser scanning confocal microscope (LSM 710, Carl Zeiss, Germany) with a C-Apochromat 40X/1.2 water immersion lens, employing a 488 nm argon laser (505–550 nm detection range for GFP) and a 561 nm solid state laser (586-662 nm detection range for PE). Post co-culture with tumor spheroids, GAS-NK cells were analyzed using a ZOE fluorescent imager microscope (BioRad, United States), and tissues stained with H&E on glass slides were examined using a Leica LMD6 system (Buffalo Grove, IL, USA).

### Flow cytometry analysis
For cellular analysis, live NK-92 cells were stained with CD45-PE antibody (Cat. A07783; Beckman Coulter, Brea, CA, USA) in PBS containing 1% FBS for 15 minutes. Target cancer cells, cultured overnight in 6-well plates, were co-cultured with NK cells for 4 hours. Post co-culture, cells were collected, resuspended in 400 µL of PBS with 0.5% FBS, and the activation of GAS-NK cells was quantified using GFP expression via flow cytometry (CytoFlex, Beckman Coulter). Data analysis was conducted using Kaluza software (Beckman Coulter, USA). For lung tissue cell analysis, SCID mouse lung tissues were dissected and homogenized at 4 °C in RIPA buffer (20188, Sigma) containing 1 mg/ml collagenase A (10103586001, Roche), 1 mg/ml hyaluronidase (H0164, TCI), and 20 µg/ml DNase I (11284932001, Sigma). After adding Proteinase Inhibitor Cocktail (1:200, P8340, Sigma), the mixture was incubated in a shaker incubator at 37 °C for 60 minutes. The homogenate was then centrifuged at 300 × g for 5 minutes, resuspended in PBS, and passed through a 40 µm cell strainer. Subsequently, the lung tissue homogenate underwent a further 5-minute incubation at 37 °C with red blood cell lysis buffer, followed by centrifugation. The homogenate was then passed through a cell filter twice and washed with PBS. For FACS analysis, NK cells were counted using a human CD56 antibody (1:100) staining.

### Reporter gene assay
Luciferase activity in cells was measured using the Luciferase Assay System kit (Promega, Madison, WI, USA), according to the manufacturer's instructions. This was performed on an M4 molecular device spectrophotometer. Cells were washed once with phosphate-buffered saline (PBS) and lysed with 100 µL of lysis buffer. The lysates were then centrifuged (4 °C, 20,000 × g, 5 min). For the assay, cell extracts were diluted 1:1 with luciferase reagent (Promega), incubated at room temperature for 10 minutes within 2 hours of preparation, and the luminescence measured. All assays were conducted in triplicate.

## In vivo imaging analysis

Six-week-old male mice were acquired from Nara Biotech (Seoul, Korea). All animal handling and care procedures strictly adhered to the guidelines set by the Institutional Animal Care and Use Committee (KBSI-IACUC-22-12) and conformed to the "Guide for the Care and Use of Laboratory Animals" as outlined by the KBSI Laboratory Animal Resources Committee. For each experimental group, a sample size of 4 to 6 mice was used. Three distinct bioluminescence imaging (BLI) experiments were conducted, assessing GAS-NK cell activation, in vivo activation capacity, and targeted homing at tumor sites. Initially, to evaluate the feasibility of monitoring GAS-NK cell activation in vivo, both normal GAS-NK cells (serving as a negative control, $2 \times 10^6$ cells) and IFNα-activated cells were intravenously injected into the tail veins of mice. Images were captured at 0 and 2.5 hours post-administration of D-luciferin (150 mg/kg, PerkinElmer, MA, USA). Secondly, to assess the in vivo activation capacity of GAS-NK cells, $2 \times 10^6$ cells suspended in PBS were administered subcutaneously into the flanks of BALB/c nude mice. Concurrently, IFNα and PBS were also administered subcutaneously into the abdomens of the mice. BLI was conducted four hours later to assess activation. Thirdly, to explore the targeted homing and activation of GAS-NK cells at tumor sites, A549 cells were injected intravenously into SCID mice to establish a tumor metastasis model. MR (9.4 T animal MRI system, Bruker Biospin, Ettlingen, Germany) imaging was performed 18 days post-tumor cell inoculation to evaluate the metastatic tumor burden. On day 20, $5 \times 10^6$ GAS-NK cells were injected via the tail vein into both the normal and metastatic animal models, followed by BLI. Mice were euthanized 24 hours later (21 days post-tumor inoculation) for ex vivo imaging analysis. The lungs were harvested to examine the homing effect of the GAS-NK cells.

## Immunoblot assay

Cells were lysed using a lysis buffer composed of 120 mM NaCl, 40 mM Tris-HCl (pH 8), and 0.1% NP-40 (Nonidet P-40), supplemented with a Protease Inhibitor Cocktail Tablet (Roche). The protein concentrations in the lysates were determined using the Bradford method, with bovine serum albumin (BSA) serving as the standard. For electrophoresis, 40 μg of each protein sample was loaded onto 10% SDS-PAGE gels. Following electrophoresis, proteins were transferred to polyvinylidene fluoride (PVDF) membranes using an iBlot dry blotting system (Invitrogen). The membranes were then blocked using the blocking buffer provided by the iBind Western System (Thermo Fisher Scientific, Inc.). Primary antibodies were applied for probing, including monoclonal antibodies for STAT1 (1:1,000, #14994S; Cell Signaling Technology), Phospho-Stat1-S727 (P-Stat1-S727, 1:1,000, #8826S; Cell Signaling Technology), Phospho-Stat1-Y701 (P-Stat1-Y701, 1:1,000, #9167S; Cell Signaling Technology), and β-actin (1:1000, sc-47778; Santa Cruz Biotechnology, Inc.). The membranes were incubated with horseradish peroxidase-conjugated secondary antibodies specific for mouse (1:200; # GTX213111-01, GeneTex) and rabbit (1:200; # GTX213110-01, GeneTex). The membranes were placed in the iBind device, and solutions, including the antibodies, were added to designated wells. The device facilitated sequential flow of these solutions across the membrane. Antibody binding was detected using enhanced chemiluminescence assays, and protein bands were visualized using a ChemiDoc™ XRS+ imaging system (Bio-Rad, Hercules, CA, USA).

## Cytokine antibody array

The cytokine antibody array was conducted using the Proteome Profiler Human XL Cytokine Array Kit (ARY022B, R&D Systems, USA). Culture mediums from GAS-NK cells, both with and without IFNα treatment, as well as supernatants from co-cultured cancer cell lines (MDA-MB-231, HCT116, HepG2, and A549), were applied to the array membrane and incubated at 4 °C overnight. The membrane was then loaded into the iBind device, and the manufacturer-supplied antibody solution was added to the designated wells. This process allowed for the sequential flow of solutions across the membrane. Protein bands were visualized using chemiluminescence analysis with a ChemiDoc™ imaging system.

## ELISA assay

Cell culture supernatants were centrifuged at $400 \times g$ for 10 min and then at $20,000 \times g$ for another 20 min. IFNα production was measured using a VeriKine Human IFNα ELISA Kit (R&D Systems) according to the manufacturer's instructions.

## RNA sequencing and data analysis

For mRNA profiling by RNA sequencing, GAS-NK cells and IFNα-treated GAS-NK cells (24-hour treatment) were used. For total RNA was extracted, cells were lysed with TRIzol reagent, followed by the addition of 0.2 mL of chloroform and incubation at −20 °C for 20 min to extract total RNA. After centrifugation at $12,000 \times g$ for 20 min, the aqueous phase containing the total RNA was obtained. The total RNA was then precipitated through the addition of 0.6 mL of isopropyl alcohol. After incubation on ice for 10 min and centrifugation at $12,000 \times g$ for 10 min, the resulting pellet was washed with 70% ethanol. Finally, the concentration of total RNA dissolved in DNase-free and RNase-free water (Invitrogen, Waltham, MA, USA) was measured using a NanoDrop™ 2000/2000c Spectrophotometer (Thermo Fisher Scientific, Waltham, MA, USA). First-strand cDNA was synthesized from 5 μg of the total RNA template through reverse transcription using oligo-dT and the SuperScript™ III First-Strand Synthesis System (Invitrogen). RNA sequencing was performed by ATG Lifetech Inc. (Seoul, Republic of Korea). All the experiments were done by triplicate, making up a total of 6 samples. Gene set enrichment analysis was completed using GSEA_Linux_4.0.3 (available at GSEA). Differential gene expression analysis was conducted using DESeq2 (version 1.4.5), with a Q value ≤ 0.05. Heatmaps were generated using pheatmap (version 1.0.8). Enrichment analysis of annotated differentially expressed genes was performed with Enricher (available at Enricher) based on a hypergeometric test.

## Statistics and reproducibility

Data collection and analysis were performed using GraphPad Prism version 9.4.1. All data are expressed as the mean ± standard error of the mean (SEM). Statistical analyses were conducted using the Student's $t$-test for normally distributed data and the Mann-Whitney U-test for data that did not follow a normal distribution. Statistical significance levels are represented as follows: ns (not significant), $P \geq 0.05$; $*P < 0.05$; $**P < 0.01$; $***P < 0.001$; $****P < 0.0001$; and $*****P < 0.00001$. Experiments were conducted with sample sizes similar to those commonly used in the field, but sample sizes were not predetermined using formal statistical methods. Each experiment was performed in at least two independent biological replicates, with technical replicates included where applicable. Replicates were defined as independently conducted experiments or measurements performed under identical conditions. Details on the exact number of replicates for each experiment are provided in the respective figure legends.

## Data availability

The numerical source data for the graphs and charts presented in this study have been deposited in Figshare and are accessible via the following [https://doi.org/10.6084/m9.figshare.26630452]. The RNA sequencing data have been deposited, and the access code is PRJCA029184. The plasmid generated in this study has been deposited in Addgene and can be accessed using the deposit number 84722 and 84726.

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

## Acknowledgements

We would like to acknowledge the generous financial support provided by the following grants. Dr. Eun Hee Han: We acknowledge the financial support received from grants NRF-2020R1C1C1012542 from the Korean Ministry of Science and Information Technology (MSIT), CCL22061-100 from the National Research Council of Science & Technology (NST) of the National Research Foundation of Korea (NRF), and E0210203 from the Korea Food Research Institute (KFRI). Dr. Kwan Soo Hong: This work was supported by the grant CAP-18-02-KRIBB from the National Research Council of Science & Technology (NST) of the National Research Foundation of Korea (NRF), and the grant A423100 from the Korea Basic Science Institute (KBSI). We are grateful for the financial resources provided by these grants, which have been instrumental in facilitating our research and advancing our understanding of NK cell therapy.

## Author contributions

K.S.H. and E.H.H. conceived and developed the ideas and concepts for this study. E.H.H. and J.Y.M. contributed to writing and developing the thesis. J.Y.M., H.M.K., H.S.L., M.Y.C., H.S.P., S.-Y.L., M.S.P., S.K.H., D.K., H.G.J., T.-D. K., K.S.H., and E.H.H. contributed to the design of the experiments, data analysis, and critical revision of the manuscript. H.S.L. and M.Y.C. performed the animal experiments. J.Y.M., H.M.K., H.S.L., M.Y.C., H.S.P., S.-Y.L., M.S.P., S.K.H., D.K., H.G.J., and T.-D.K. actively participated in the in vitro 2D and 3D experiments.

## Competing interests

The authors declare no competing interests.

## Ethics

The animal study protocol was reviewed and approved by the Committee of the Korean Basic Science Institute (KBSI-IACUC-22-12).
