## [Peer Review File · Communications Biology]

This manuscript has been previously reviewed at another *Nature Portfolio* journal. This document only contains reviewer comments and rebuttal letters for versions considered at *Communications Biology*.

Reviewers' comments:

Reviewer #1 (Remarks to the Author):

In the submitted manuscript by Min et al, the authors presents a non-invasive and selective method to monitor NK cells activation by engineering NK cells to incorporate an IFN α activation sequence which can be used as an indicator for STAT1 activation. Moreover, the authors showed comprehensive data to validate the efficiency of the model including both in vivo and in vitro which is impressive.

The question I hold for the authors is that the reason to have IFN α -STAT1 signaling as a reporter for NK cell activation as there are tons of signaling pathway activated upon NK cells activation. How about IFN gamma or GzmB? What is the advantage of the model over others?

The Line #26, is it IFN γ or IFN α ? Please clarify.

In Figure 1, the authors detected peak level at 24h for the luciferase assay, how about dynamics of IFN α level? Is there any correlation?

Reviewer #2 (Remarks to the Author):

Title: Non-Invasive Monitoring of NK Cell Response in Cancer: A Focus on STAT1 Activation

Overall: Jin Young Min et al. have developed a new methodology to directly follow and in real-time NK cell by non-invasive imaging modalities. To do that, the authors, engineered NK cells to incorporate an IFN gamma activation sequence with a reporter, which binds to the STAT1 transcription factor. This cell named in the manuscript GAS-NK cell, have shown a high NK cell activity STAT1 dependent compared to NK cell and according to the tumor cell lines. The authors were able to observe in vivo, in the mice model the GAS-NK cell localization and activity by bioluminescence detection. The authors have also shown in vitro experiments, that the GAS-NK activity can be restricted by TGF beta production. The study submitted by the authors is interesting and highlights an important objective for the development of future NK cell therapy. However, there are some points that need to be clarified in the manuscript.

Major comments:

- A lot of mistakes concerning to the figure annotation in the manuscript. It is important to edit that.

- The section mentioning alpha PDL1-CAR-GAS-NK cells does not have any figure associated in the manuscript.
- The NK cell use for the GAS-NK cell is NK-92. Because the study does not present humanized mice, the limitation of the GAS-NK cell activity can be important. Despite A549 cells are human cell line the lung environment could be mismatching concerning to NK cell activity support. So, for more in vivo relevance, did the authors use a lung tumor mice model with GAS-NK cell from mice NK cells or work in humanized mice?
- In NK cell therapy, the efficient NK cell lines are irradiated before injection to avoid a loss of control of NK cell response. Did the authors irradiate the GAS-NK cells? If yes, does the treatment impact the GAS-NK cell activity?
- In the figure 5d, the % of GFP(+) population at 24 hours for GAS-NK:A549 is at the same level than K562 level represented in the figure 3d and 4a. However, the K562 is presented as stimulating condition for GAS-NK cell. While the IFN alpha treatment is at the same level in all figures. Can the authors increase the experiment to figure out more precisely the variation range for these three conditions and confirm the significant differences between IFN alpha treatment, K562, and A549 cell line?
- The authors have measured the GAS-NK cell activity with CD107a. To confirm the effect of GAS-NK cell in vivo, did the authors measure the lung inflammation in the lung normal versus tumor.
- The purpose of the study is to develop a new imaging method for NK cell in vivo. In vitro, the pic of activity was found after 24h. However, in vivo condition and in vitro condition are particularly different. Did the authors measure in kinetic (days) to measure how long the GAS-NK cell signal can be followed in the mice after injection.
- In supplemental Figure 3, the gating strategy need to show a NK marker such as CD56, particularly concerning to the "Control" that is GFP negative where there is not any marker to confirm the cells showed are NK cells.
- Did the authors calculate the z-score of the K562 "binding of STAT binding site " (supplementary figure 5d)?
- Interestingly, TGF beta is described by the authors as a factor limiting the GAS-NK cell activity. Do the authors measure the TGF beta receptor expression at the surface of GAS-NK cells?
- Please mention in the legend of the figures the number biological replicates used.

Minor comments:

- Consistency about "non-invasive" instead of "noninvasive"
- Consistency about "g" instead of "rpm"
- Detail the method at the line 256: " Total RNA was extracted, and a cDNA library was prepared"

- In addition to correct the figure annotation in the text, change the order of the figure to follow the text.

- The ChIP was done after 24h of IFN alpha treatment?

- The GAS-NK cell described an important increase of the NK cell activity STAT1 dependent in IFN alpha treatment. However, the comparison GAS-NK IFN alpha treatment versus no IFN alpha treatment do not seem the best to elucidate and confirm the engineer NK cell benefice STAT1 dependent. Indeed, did the authors compare the GAS-NK activity in IFN alpha treatment versus NK activity in IFN alpha treatment?

- Did the authors see a better overall survival mouse with GAS-NK cell injection compare to NK cell injection?

Rebuttal Letter

Manuscript Title: Non-Invasive Monitoring of NK Cell Response in Cancer: A Focus on STAT1 Activation

Manuscript ID: COMMSBIO-24-1792-T

Reviewer #1:

Comment 1: The rationale for using IFN α -STAT1 signaling as a reporter for NK cell activation instead of other pathways such as IFN γ or Granzyme B.

Response: We selected STAT1 due to its specific role in regulating NK cell development, IFN- γ production, and cytotoxic capacity, particularly within the tumor microenvironment. This choice enhances the accuracy of monitoring NK cell responses to tumor cells. We have detailed this rationale in the Discussion section.

Comment 2: Clarification of the IFN γ activation sequence on line #26.

Response: We clarified that the GAS element refers to the IFN γ activation sequence, with STAT1 transcriptional regulators binding to the GAS element and inducing IFN γ production. This clarification is now included in the manuscript.

Comment 3: Correlation between IFN α levels and luciferase activity over time.

Response: We conducted time-course studies showing that while luciferase activity peaks at 24 hours and declines after 48 hours, IFN γ levels remain elevated until 72 hours. These results, now included in Figure S1g, highlight the differences in dynamics between the reporter protein and endogenous cytokines.

Reviewer #2:

Comment 1: Correction of figure annotations and consistency in the manuscript.

Response: We have meticulously revised the figure annotations and ensured consistency throughout the manuscript. This improves the clarity and accuracy of our presentation.

Comment 2: Integration of α PDL1-CAR-GAS-NK cells into the manuscript with associated figures.

Response: We revised the manuscript to include data on α PDL1-CAR-GAS-NK cells, demonstrating their functionality and activity. These results are now presented in Supplementary Figure 10.

Comment 3: Justification for using the NOD/SCID mouse model instead of humanized mice.

Response: We provided a detailed explanation for choosing the NOD/SCID model, highlighting its suitability for *in vivo* monitoring and evaluating antitumor efficacy. We discussed the limitations of humanized mouse models and supported our choice with references.

Comment 4: Impact of irradiation on GAS-NK cell activity.

Response: We clarified that we did not perform lethal irradiation to monitor GAS-NK cell activation and distribution *in vivo*. Previous studies support our decision, as GAS-NK cells did not exhibit uncontrolled proliferation or tumorigenicity within the monitoring period. Our primary goal was to monitor the activation of GAS-NK cells *in vivo* and track their distribution in real-time. The NK-92 cell line, which we used, has a doubling time of 24–36 hours. To effectively monitor *in vivo* distribution of GAS-NK cells beyond this period, we decided not to perform lethal irradiation on the cells.

Comment 5: Verification of the correlation between GAS-NK activity and tumor cell lines.

Response: We consolidated experimental groups into a single visual representation (Supplementary Figure 6d) and confirmed significant differences in GFP activity across various conditions, reinforcing our findings.

Comment 6: Measurement of lung inflammation in normal versus tumor tissues.

Response: We analyzed TNF- α and IL-6 expression levels in adjacent normal and tumor tissues, finding that TNF- α levels remained unchanged while IL-6 was significantly elevated in tumor tissues. These results are included in Supplementary Figure 7d.

Comment 7: Differences between in vitro and in vivo conditions.

Response: We conducted time-course studies to monitor GAS-NK cell activity in vivo, revealing rapid initial activity followed by a decline. These findings, now included in Figure S8a, highlight the dynamic response in the in vivo environment.

Comment 8: Incorporation of NK cell markers in gating strategy.

Response: We explained our rationale for using CD45 instead of CD56 and provided additional data supporting the accurate identification of NK cells. These revisions are included in Supplementary Figure 3a.

Comment 9: Comparison of GAS-NK activity with and without IFN α treatment.

Response: We added new data comparing the cytotoxicity of empty-NK and GAS-NK cells under IFN α treatment, showing no significant difference. This comparison is included in Supplementary Figure 1c.

Comment 10: Overall survival analysis with GAS-NK cell injection.

Response: We revised the manuscript to include findings on overall survival, showing no significant improvement with GAS-NK cells compared to empty-NK cells. These results are presented in Supplementary Figure 8b.

We hope that these revisions adequately address the reviewers' concerns and improve the manuscript's quality. We look forward to your feedback and the opportunity for further review.

Thank you for considering our resubmission.

Sincerely,

Eun Hee Han, Ph.D.

Biopharmaceutical Research Center,

Ochang Institute of Biological and Environmental Science,

Korea Basic Science Institute, Korea

Email: heh4285@kbsi.re.kr

REVIEWERS' COMMENTS:

Reviewer #1 (Remarks to the Author):

I appreciate the authors' point-to-point response to my previous comments. I look forward to the final version of the manuscript. Congrats!